# Cryo-EM structure of *Helicobacter pylori* urease with an inhibitor in the active site at 2.0 Å resolution

Eva S. Cunha [1 ✉], Xiaorui Chen [2,6], Marta Sanz-Gaitero [1], Deryck J. Mills[3] & Hartmut Luecke [1,2,4,5 ✉]

Infection of the human stomach by *Helicobacter pylori* remains a worldwide problem and greatly contributes to peptic ulcer disease and gastric cancer. Without active intervention approximately 50% of the world population will continue to be infected with this gastric pathogen. Current eradication, called triple therapy, entails a proton-pump inhibitor and two broadband antibiotics, however resistance to either clarithromycin or metronidazole is greater than 25% and rising. Therefore, there is an urgent need for a targeted, high-specificity eradication drug. Gastric infection by *H. pylori* depends on the expression of a nickel-dependent urease in the cytoplasm of the bacteria. Here, we report the 2.0 Å resolution structure of the 1.1 MDa urease in complex with an inhibitor by cryo-electron microscopy and compare it to a β-mercaptoethanol-inhibited structure at 2.5 Å resolution. The structural information is of sufficient detail to aid in the development of inhibitors with high specificity and affinity.

[1] Structural Biology and Drug Discovery Group, Centre for Molecular Medicine Norway, Nordic EMBL Partnership, University of Oslo and Oslo University Hospital, 0318 Oslo, Norway. [2] Department of Molecular Biology and Biochemistry, University of California, Irvine, CA 92697, USA. [3] Department of Structural Biology, Max Planck Institute of Biophysics, Frankfurt am Main, Germany. [4] Department of Medical Biochemistry, University of Oslo and Oslo University Hospital, 0372 Oslo, Norway. [5] Department of Physiology and Biophysics, University of California, Irvine, CA 92697, USA. [6]Present address: Genomics Research Center, Academia Sinica, 128 Academia Road, Sect. 2, Nankang District, Taipei, Taiwan. ✉email: cunha@uio.no; hudel@uio.no

*H*elicobacter pylori is a Gram-negative neutralophile that has acquired a set of genes called the *ure* gene cluster, that, in the presence of urea, enable the bacterium to survive at extremely acidic pH. Exploiting this unique ability, *H. pylori* is estimated to be colonizing the stomachs of roughly half the world population, causing a wide spectrum of diseases ranging from gastritis and gastric ulcers to stomach cancer[1,2]. Gastric cancer is the third most common cause of cancer death worldwide and more than 90% of the cases are attributable to chronic *H. pylori* infection[3].

Current eradication, called triple therapy, entails ingesting a proton-pump inhibitor and two broadband antibiotics, however, resistance to antibiotics clarithromycin and metronidazole is generally greater than 25% and rising[4]. This resistance has resulted in *H. pylori*'s inclusion in the WHO Global Priority List of Antibiotic-Resistant Bacteria[5,6], underscoring the urgent need for a targeted, high-specificity *H. pylori* eradication drug.

The *ure* gene cluster is comprises seven genes, two of which code for a nickel-dependent urease (*ureA* and *ureB*) that hydrolyzes urea into $NH_3$ and $CO_2$, one for a pH-gated urea channel (*ureI*) that delivers host gastric urea to urease in the bacterial cytoplasm, and four for cytoplasmic accessory proteins involved in nickel processing. In addition, other proteins, such as a periplasmic α-carbonic anhydrase, are also required for colonization of the human stomach[7].

*H. pylori*'s proton-gated plasma membrane urea channel[8] and cytoplasmic urease are both virulence factors and essential for its survival in the stomach[9]. The channel senses periplasmic pH and, while closed at neutral pH, opens upon acidification to allow diffusion of urea from the gastric juice to the cytoplasmic urease, which comprises about 10% of the total bacterial protein[10]. Once in the cytoplasm, urea is hydrolyzed rapidly by urease into ammonia and carbon dioxide, buffering the cytoplasm and periplasm even in gastric juice at acidity levels below pH 2.

Previously, we determined the crystal structure of the UreI urea channel, revealing six protomers assembled in a $C_6$ hexameric ring surrounding a central bilayer plug of ordered lipids[11]. The channel architecture coupled with unrestrained all-atom molecular dynamics studies suggested a mechanism for low-flux urea passage ($\sim 10^4$ molecules channel$^{-1}$ s$^{-1}$), as we well as high-flux water passage ($\sim 8 \times 10^9$ molecules channel$^{-1}$ s$^{-1}$)[12]. More recently, the cryo-electron microscopy (cryo-EM) structures of the channel at acidic and neutral pH revealed structural details of the pH gating mechanism[13].

Ureases (EC 3.5.1.5) are amidohydrolases found in bacteria, algae, plants and fungi with an active site composed of a carbamylated lysine (KCX)[14] coordinating a bi-nickel center. In rare variants the active site contains two iron cations instead of nickel, presumably to overcome low-nickel conditions, yielding a less active enzyme[15]. In the 1920s, jack bean urease was the first enzyme to be crystallized[16], however, it took until 1995 for the first three-dimensional (3D) urease structure to be reported, that belonged to *Klebsiella aerogenes*[17]. Ureases form stable $C_3$ trimers, although some occur in higher-order arrangements, either as $D_3$ dimers of trimers or as tetrahedral (*T*) tetramers of trimers. In the native state of the enzyme, a hydroxide anion has been reported to bridge the two nickel ions, primed for nucleophilic attack on the substrate during hydrolysis[18–20].

Numerous urease inhibitors have been reported, that can both suppress growth of various human pathogens, as well as inhibit soil ureases that cause environmental and economic damage when urea is used as a fertilizer[21]. The three main classes are (1) sulfhydryl compounds, including β-mercaptoethanol (BME), (2) hydroxamic acid derivatives such as acetohydroxamic acid (AHA), and (3) amides and esters of phosphoric acid, such as fluorofamide, which are thought to represent transition state analogs. Fluorofamide showed some promise in animal models but was unable to eradicate *H. mustelae* in ferrets, presumably due to instability of the compound under acidic conditions[22]. We identified a set of *H. pylori* urease inhibitors using in vitro high-throughput screening (HTS) of a diverse library of ~200,000 compounds, nearly all of which turned out to be hydroxamic acid derivatives (manuscript in preparation).

For the urease from *H. pylori*, two crystal structures of the native and the AHA-bound form, both at 3.0 Å resolution, showed a dodecameric arrangement with a variable flap covering the active site[23]. Here, we determine the structure of the 1.1 MDa dodecameric *H. pylori* urease in complex with an inhibitor derived from HTS to a resolution of 2.0 Å using cryo-EM. With fewer than ten cryo-EM structures of unique complexes at a resolution of 2 Å or better, use of this technique in structure-guided drug development is still rare.

## Results

**Map quality and overall arrangement of urease complex**. We report cryo-EM maps of *H. pylori* urease at 2.5 Å and 2.0 Å resolution, the highest resolution to date for *H. pylori* urease, of sufficient detail to aid in drug development (Fig. 1 and Supplementary Figs. 1 and 2). The map at 2.5 Å resolution depicts urease with BME bound in the active site (U-BME) whereas the map at 2.0 Å details the binding of an inhibitor 2-{[1-(3,5-dimethylphenyl)-1H-imidazol-2-yl]sulfanyl}-N-hydroxyacetamide (U-SHA). Briefly, we used the program Relion[24] to obtain maps of U-BME with a resolution of 2.55 Å and of U-SHA with a resolution of 2.09 Å. Further map processing using the recently published Phenix Resolve density modification algorithm[25] improved map quality, as well as the nominal resolution to 2.5 Å and 2.0 Å for U-BME and U-SHA, respectively (Table 1 and Supplementary Fig. 1). Local resolution estimates using the program Resmap[26] show that the vast majority of the density is at the nominal resolution, while only solvent-exposed areas on the outside surface show more variability with the lowest resolution estimates around 3.1 Å for U-SHA (Supplementary Fig. 3).

Based on the two maps, using iterative model fitting and refinement, we derived atomic models for U-BME and U-SHA (Table 1). The overall single-particle cryo-EM structures of *H. pylori* urease reveal a 1.1 MDa dodecameric complex of 12 UreA and 12 UreB subunits in a tetrahedral arrangement, in agreement with the previously published crystal structures (Figs. 1 and 2)[23].

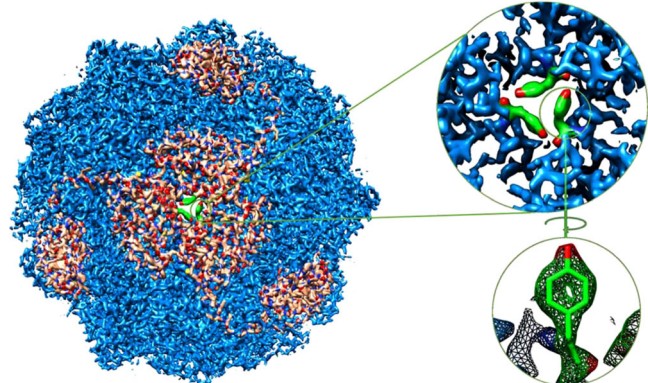

**Fig. 1 Cryo-EM density map at 2.0 Å resolution of dodecameric 1.1 MDa *Helicobacter pylori* urease with bound SHA.** Cryo-EM density for copies of UreA (regulatory subunit) are highlighted in light brown and density for tyrosine residues at a threefold axis are highlighted in green (left). Enlarged view of density along a threefold axis of the dodecameric arrangement depicting Tyr15 of UreA (top right). Cryo-EM density (mesh) and final model (solid) of Tyr15 side chain (bottom right).

Our urease models are free of Ramachandran outliers (Table 1), whereas the published 3.0 Å crystal structures of native and AHA-bound *H. pylori* urease contain 7.1% and 3.9% outliers, respectively. To evaluate the effect of modern refinement programs on crystal structure quality, we re-refined the previously published crystal structures with the Phenix package including Ramachandran restraints[27], reducing the fraction of Ramachandran outliers of the crystal structures to 3.9% and 1.6%, respectively.

The densely packed spherical complex formed by the tetragonal arrangement of urease trimers has a diameter of approximately 160 Å (Figs. 1a and 2). Each trimer is stabilized by extensive contacts between three copies of UreA and three copies of UreB with a buried surface area of over 20,000 Å$^2$ per trimer (Fig. 2). In contrast, contacts between trimers are far less extensive with a buried surface area of ~2400 Å$^2$ per trimer. It is not yet clear why certain ureases, those of *H. pylori* and *H. mustelae* included, form spherical dodecameric supercomplexes, while most known ureases occur as trimers or dimers of trimers. One hypothesis posits that generation of ammonia in the cytoplasm needs to be regulated tightly to avoid damage of the bacteria. This is achieved partially by pH gating of the urea channel responsible for delivering urea[8]. In addition, spherical dodecamers of urease docked at or near the plasma membrane[28] in which the urea channel hexamers are embedded might function in directing the products of urea hydrolysis retrograde to the periplasm, where ammonia and bicarbonate are required for effective buffering in an acidic environment.

A mutational analysis of residues at the inter-trimer interface would elucidate whether *H. pylori* urease is enzymatically active as a trimer in vitro, and also whether trimers are sufficient for acid survival in vivo. Such a study would indicate if it might be useful to develop drugs that disrupt dodecamer formation by preventing inter-trimer contacts.

**Comparison between U-BME and U-SHA and flap motion.** *H. pylori* urease is inhibited by both BME and SHA. We determined urease IC$_{50}$ values under identical conditions for BME and SHA as 13.5 mM and 19.6 μM, respectively (Fig. 3). Using these IC50 values and a substrate (urea) concentration of 6 mM coupled with the reported $K_m$ of *H. pylori* urease of 0.2 mM[29], the Cheng-Prusoff equation[30] for competitive inhibition yields $K_i$ values of 435 μM for BME and 0.630 μM for SHA, demonstrating that SHA is a much higher affinity inhibitor than BME.

The structures of *H. pylori* urease with bound BME and SHA display the same overall fold, with low backbone variations for subunit UreA (below 1 Å), although the C-terminal residues A228-238 show slightly higher root mean square deviation (RMSD) values (below 1.5 Å) (Fig. 4a, b). An analysis of the intra- and inter-trimer interfaces using the program PISA[31] reveals that the C-terminal residues A235–237 of UreA are involved in the interface between UreA and UreB subunits in inter-trimer interactions. This suggests that the presence of the SHA inhibitor influences the inter-trimer interface through conformational changes at the C terminus of the UreA subunit. The catalytic subunit UreB also shows similar low backbone variations, except for the flap region (residues B310–345), a helix–loop–helix structure in direct contact with the active site that is important for catalysis[32,33] (Fig. 4c, d). The flap region shows the highest RMSD backbone variations of both UreA and UreB subunits (above 1.5 Å), whereas the active site residues nearby exhibit very low RMSD between U-BME and U-SHA (Fig. 4b).

BME coordinates to the two Ni$^{2+}$ ions through its sulfur (Fig. 5 and Supplementary Fig. 4) in a fashion similar as described for

| | U-BME (EMDB-4629) (PDB 6QSU) | U-SHA (EMDB-11233) (PDB 6ZJA) |
|---|---|---|
| **Table 1 Cryo-EM data collection, refinement, and validation statistics.** | | |
| Data collection and processing | | |
| Magnification | 130,000 | 165,000 |
| Voltage (kV) | 300 | 300 |
| Electron exposure (e$^-$/Å$^2$) | 40 | 40 |
| Defocus range (μm) | −0.5 to −3.0 | −0.5 to −3.0 |
| Pixel size (Å) | 1.077 | 0.8426 |
| Symmetry imposed | T | T |
| Initial particle images (no.) | 193,289 | 276,994 |
| Final particle images (no.) | 175,895 | 187,461 |
| Map resolution (Å) | 2.5 | 2.0 |
| FSC threshold | 0.143 | 0.143 |
| Refinement | | |
| Initial model used (PDB code) | 1E9Z | 6QSU |
| Model resolution (Å) | 3.0 | 2.5 |
| FSC threshold | NA | 0.143 |
| R.m.s. deviations | | |
| Bond lengths (Å) | 0.006 | 0.005 |
| Bond angles (°) | 0.633 | 0.857 |
| Validation | | |
| MolProbity score | 1.55 | 1.09 |
| Clashscore | 4.57 | 0.60 |
| Poor rotamers (%) | 0.61 | 1.37 |
| Ramachandran plot | | |
| Favored (%) | 95.45 | 95.85 |
| Allowed (%) | 4.55 | 4.15 |
| Disallowed (%) | 0 | 0 |

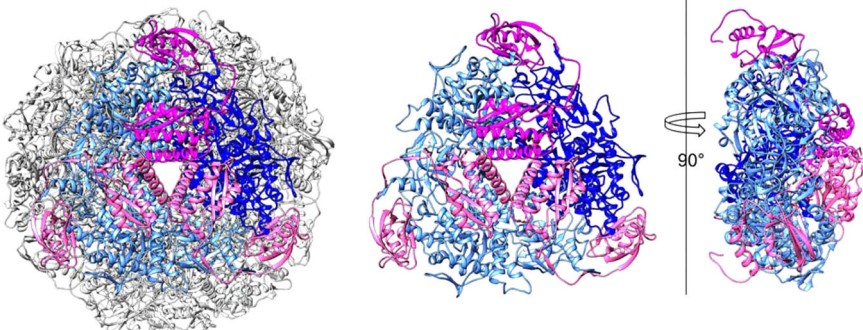

**Fig. 2 Cryo-EM model of *Helicobacter pylori* urease with one trimer highlighted.** Dodecamer where one trimer contains three copies of UreA highlighted in shades of pink and three copies of chain UreB highlighted in shades of blue (left). One trimer only using the same coloring scheme (center). Sideview of one trimer rotated 90° around a vertical axis (right).

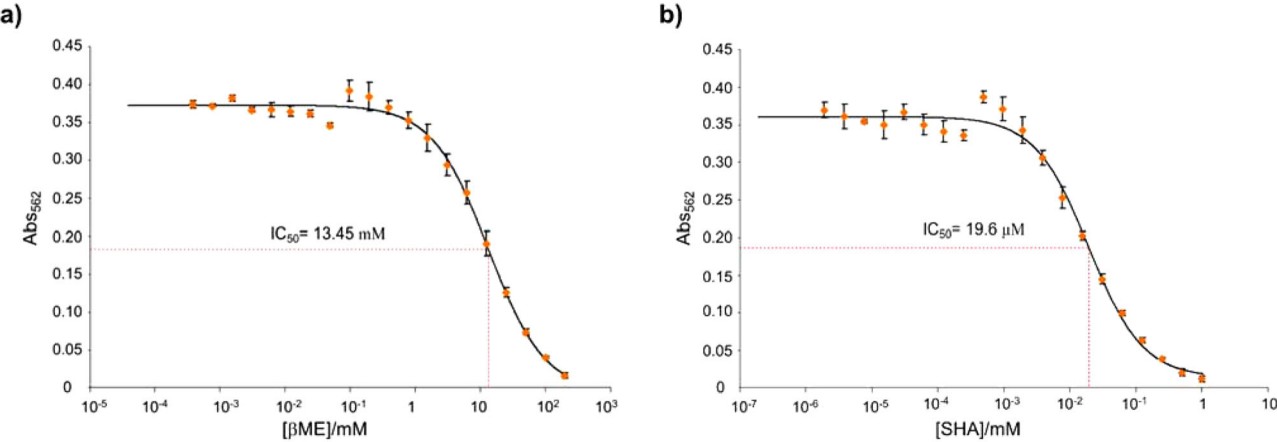

**Fig. 3 In vitro inhibition of *H. pylori* urease by BME and SHA.** Dose-response curves measuring pH increase due to ammonia generation as a function of inhibitor concentration using a phenol red spectrometric assay (absortion at 562 nm). **a** BME inhibition curve showing an $IC_{50}$ of 13.5 mM under the conditions (see "Methods"). Using this value, a substrate (urea) concentration of 6 mM and the reported $K_m$ of *H. pylori* urease of 0.2 mM[29], the Cheng-Prusoff equation[30] assuming competitive inhibition yields a $K_i$ of 435 μM for BME. **b** SHA inhibition curve with an $IC_{50}$ of 19.6 μM. The corresponding $K_i$ is 0.630 μM. Data shown were obtained from $n = 3$ independent samples in one experiment. Data are presented as mean values $+/-$ standard deviation.

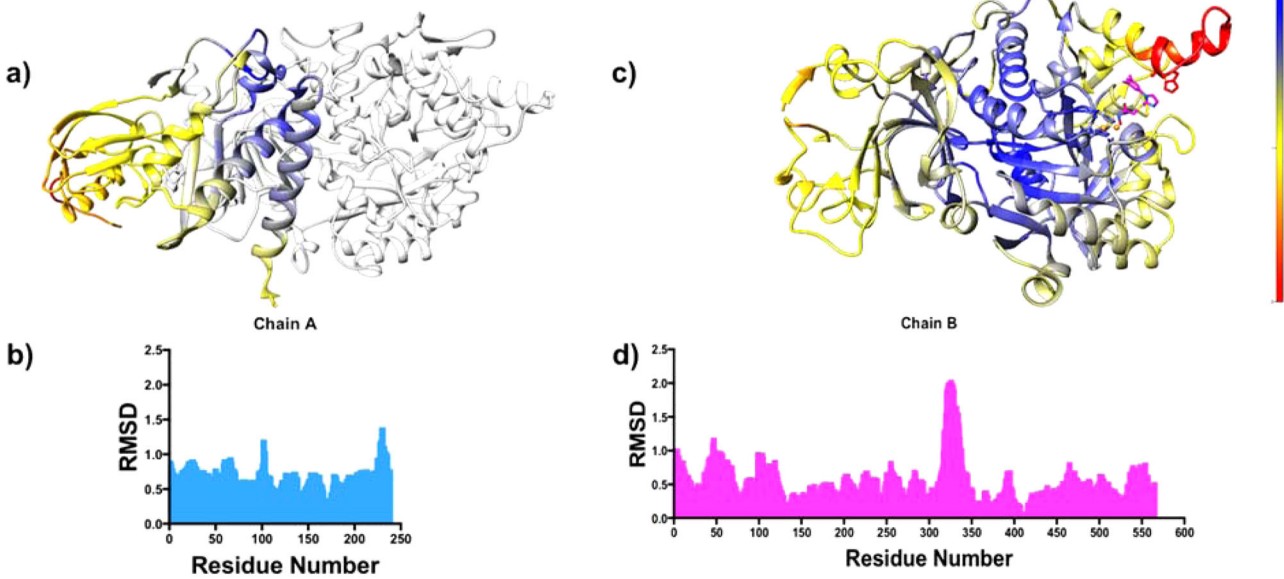

**Fig. 4 Backbone RMSD between U-BME and U-SHA. a** UreA backbone RMSD with chain B for reference in light gray (gradient of high, medium, and low RMSD values in a gradient of red, yellow and blue). **b** Histogram of UreA per residue RMSD distribution. **c** UreB backbone RMSD (gradient of high, medium, and low RMSD values in a gradient of red, yellow, and blue). **d** Histogram of UreB per residue RMSD distribution.

BME binding to *Bacillus pasteurii* urease, however, we see density for only one BME molecule, whereas *Bacillus pasteurii* urease shows density for two BME molecules[34]. In contrast, the SHA inhibitor coordinates to the two nickel ions through the hydroxyl of its hydroxamic acid moiety. Nevertheless, BME and the hydroxamic acid moiety of SHA assume similar conformations, probably driven by the requirements of $Ni^{2+}$ coordination. The difference in flap region positioning appears to be due to the bulky SHA imidazole and phenyl rings that would sterically clash with the side chains of Cys321 and His322 at the tip of the flap in the BME complex, with both residues fully conserved for the 260 GenBank deposited *H. pylori* urease sequences. This steric hindrance results in a displacement of the flap residues away from the active site, with residues UreB B321-324 having RMSD values close to 2.0 Å. Interestingly, a comparison of the flap region between the four available structures (U-BME and U-SHA herein, and the two previously published crystal structures,

U-NAT and U-AHA, the native and AHA-bound structures, respectively) shows that these structures can be clustered into three different groups according to the distance between the tip of the flap and the active site (Supplementary Fig. 5). The distance between the α carbon of His322 and the two nickel ions varies between 9–10 Å for U-NAT, 13–14 Å for U-BME as well as U-AHA, and 14–15 Å for U-SHA, indicating that the major differences between these structures occur in the flap region and between U-NAT and U-SHA, likely due to the large size of the SHA inhibitor, relative to both AHA and BME, which are similar in size.

**SHA inhibition mode.** The density for SHA shows that its hydroxamic acid moiety interacts with both active site $Ni^{+2}$ while its rings interact with the UreB flap region. The density of both rings of the SHA inhibitor is better defined than the density for

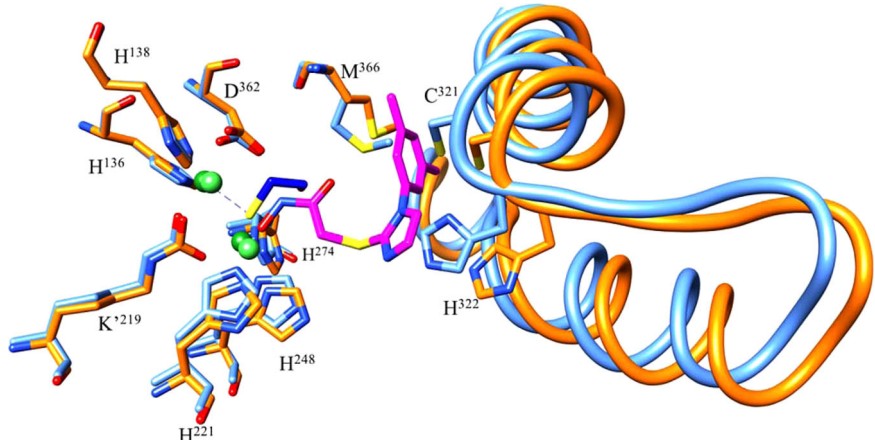

**Fig. 5 Overlap of U-BME (light blue) and U-SHA (orange) active site (sticks) and flap region (ribbon and sticks).** SHA is shown in magenta and BME in dark blue with sulfur atoms in yellow. $Ni^{2+}$ ions are depicted in green (U-SHA: light green, U-BME: dark green). K' corresponds to the carbamylated Lys219 coordinating the nickels in the active site. The side chains of flap residues Cys321 and His322 are shown.

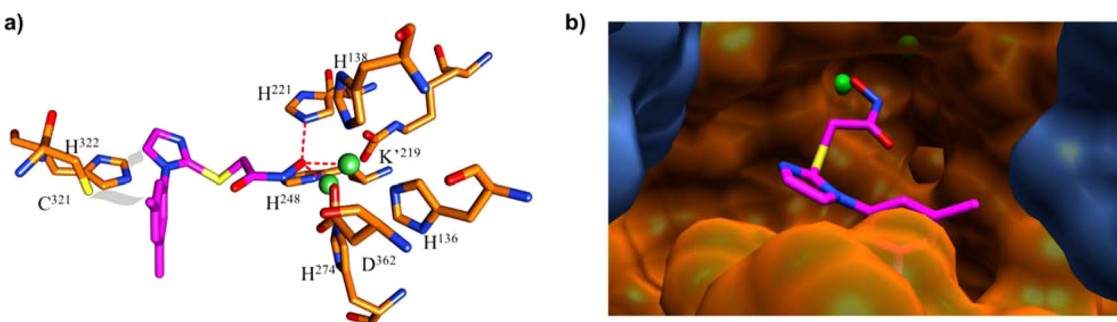

**Fig. 6 SHA bound at the active site of *H. pylori* urease. a** Interactions of SHA, with dashed red lines showing interactions of the hydroxyl moiety of the hydroxamic acid: a hydrogen bond from His221 $N_{\varepsilon 2}$ and coordination of the two $Ni^{2+}$ ions. In gray van der Waals interactions of the SHA imidazole and phenyl groups with the tip of the flap region. **b** Binding pose of SHA with the surface of catalytic subunit UreB in orange and the tip of the flap region in the foreground, while the surface of neighboring UreB' is shown in blue. The two $Ni^{2+}$ ions are shown in green. Note the space to the left of the SHA imidazole toward UreB' that could be exploited with modified inhibitors.

C14 in the chain leading to the hydroxamic acid moiety, suggesting that there is flexibility. SHA coordinates with the bi-nickel center through its hydroxamic acid hydroxyl O18 with distances of 1.98 and 3.10 Å (Fig. 6a). Since most hydroxamic acids have reported solution $pK_a$ values between 7.9 and 9[35], we speculate that hydroxamic acids in general, and SHA in particular, coordinate the two active-site $Ni^{2+}$ of *H. pylori* urease in their deprotonated, hydroxy-anion form, replacing the hydroxide sandwiched between the nickels in the native enzyme[36]. In addition, SHA O18 accepts a 3.06 Å hydrogen bond from His221 $N_{\varepsilon 2}$. Interestingly, despite the binding of a bulky inhibitor the conformation of the active site residues and the coordination between the active site residues and the bi-Ni center is not disturbed when compared to U-BME and to the previously published U-NAT and U-AHA crystal structures (Fig. 6a)[23]. Further away from the hydroxamic acid moiety, van der Waals interactions occur between the imidazole and phenyl rings of the inhibitor and flap region residues, especially conserved residues Cys321 and His322 whose side chains are within 3.5 Å of SHA. Inhibitor binding results in significant displacement of the flap region residues relative to U-BME and the U-NAT and U-AHA crystal structures.

Interestingly, another copy of a catalytic UreB subunit from the same trimer (UreB') defines parts of the active site. On one side, the main chain NH of UreB' Gly47 is located 5.6 Å from the SHA imidazole C10 (Fig. 6b). On the other side, the side chain of UreB' Ile467 is situated 3.7 Å from the C10 methyl of the SHA phenyl

group. Moreover, while the latter region is packed quite densely, future drug development efforts might be able to exploit the space extending from the SHA imidazole, especially toward the nearby conserved Cys321 (3.5 Å from SHA imidazole C9) in the flap region as well the charged Lys49 of UreB' (7.5 Å from the SHA imidazole C10).

## Discussion

*Helicobacter pylori* urease is a validated drug target for the eradication of pervasive chronic stomach infection[9] that leads to severe human health diseases such as gastritis and stomach cancer. Despite decades of research, there are no treatments that specifically target *H. pylori* machinery and the current extensive treatment protocols are increasingly failing due to resistance to broadband antibiotics, high rates of antibiotic-associated side effects and low compliance[37,38]. Our research focuses on the urgent need for the development of a more targeted eradication therapy.

Here, we report the first high-resolution structure of urease from *H. pylori* with a complex hydroxamic acid inhibitor that has the potential to open the door to structure-guided drug development using cryo-EM. Specifically, modifications to the imidazole moiety that would result in increased interactions with the tip of the flap region or with a nearby section of a neighboring UreB molecule might be able to lead to compounds with lower $IC_{50}$ values (Fig. 6a, b). Currently identified urease inhibitors such as acetohydroxamic acid and fluorofamide[39,40] have met various

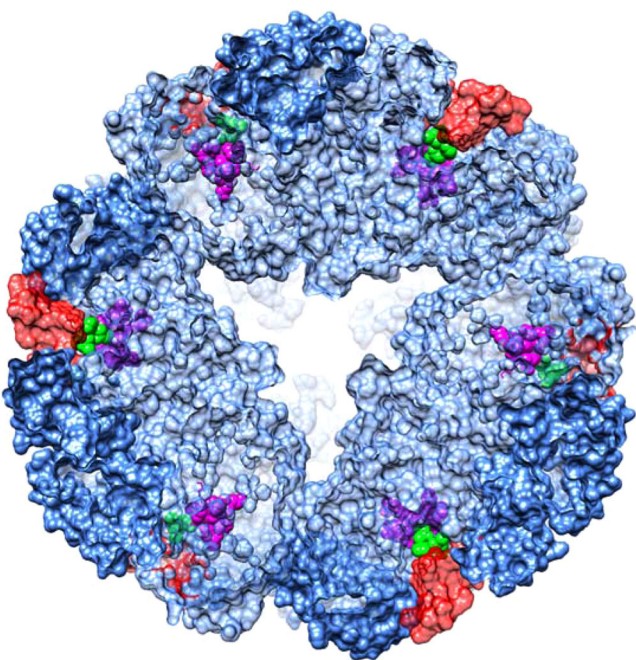

**Fig. 7 Active sites and bound SHA molecules in a 40 Å slice of the *H. pylori* urease dodecamer.** Six sets of active site residues (magenta), the flap region (red) and SHA inhibitor molecules (green) of the total of twelve sets are visible in this slice of the dodecamer. The active sites are not readily accessible from the outer surface of the dodecamer.

challenges. AHA, although approved by the Food and Drug Administration in the United States, has raised safety concerns due to severe side effects such as teratogenicity, psychoneurologic and musculo-integumentary symptoms[39], whereas fluorofamide has a short half-life at acidic pH[22]. Therefore, any newly developed compounds would need to withstand the acidic pH of the stomach and be able to cross both the outer and inner membrane of *H. pylori* to reach their cytoplasmic target, besides overcoming safety concerns.

With regards to substrate and product diffusion to and from the active site of dodecameric *H. pylori* urease it is noteworthy that for U-SHA the distance from the bi-nickel center to the outer surface is about 30 Å and presumably requires movement of the flexible flap region for access (Fig. 7). This has implications for the mechanism of how the urea substrate enters the active site, as well as for how the products, ammonia and carbon dioxide, exit the reaction vessel. U-SHA adds a structural snapshot with the most open flap region observed experimentally to date, that can be used for studying the mechanistic details of entry and exit of the substrate and products using molecular dynamics approaches[41].

Structure-based drug discovery relies on high-resolution maps and on the resulting accurate 3D models. To date, the use of cryo-EM is not yet routine for this purpose as there are only few structures with a resolution of 2 Å or better, most of which are those of reference proteins such as apoferritin and β-galactosidase. Preparation of high-quality cryo-grids as one of the major bottlenecks in cryo-EM will still have to be overcome, however, with the advent of faster direct electron detectors (such as the Gatan K3) and improved energy filters, coupled with optimized data collection and processing protocols, cryo-EM has the potential to yield 2 Å datasets in a matter of hours and refined structures in a matter of days.

## Methods

**Protein expression and purification.** The *E. coli* BL21(DE3) strain was transformed with both pHP808 and pHP902 plasmids[42], containing the entire urease gene cluster (pHP808) and only the urease structural genes *ureA* and *ureB* (pHP902), respectively, and scaled up to grow at 37 °C for 24 h in M9 salt medium with 2 mM MgSO$_4$, 1 µM NiCl$_2$, 100 µM CaCl$_2$, 0.002% thiamine and 0.4% glucose. Cells were harvested, sonicated in lysis buffer (50 mM potassium phosphate pH 6.8, 30 µg/mL DNase, 10 mM BME, 0.5 mg/mL lysozyme), and clarified by centrifugation at 27,000 × *g* for 30 min at 4 °C. The supernatant was subjected to two consecutive rounds of ammonium sulfate precipitation (50 and 80%), and *H. pylori* urease was found in the pellet of the second precipitation (80%). The re-dissolved pellet was buffer-exchanged into the low-salt buffer A for MonoQ anion exchange chromatography (20 mM potassium phosphate pH 6.8, 1 mM BME and 1 mM EDTA), and *H. pylori* urease was eluted in a gradient of buffer B (buffer A plus 500 mM KCl). Partially purified urease was then subjected to Superose 6 size-exclusion chromatography in the final buffer (20 mM Hepes pH 7.5, 20 mM NaCl and 1 mM TCEP). The final purity of *H. pylori* urease was better than 99.5% as the large size of the 1.1 MDa dodecamer caused it to elute ahead of impurities.

**Cryo-EM grid preparation and data collection.** The grids for cryo-EM were prepared according to standard procedures. In short, 3 µL of urease samples at 1.3 mg/mL with 1 mM BME or with 10 mM SHA were applied to freshly glow-discharged Quantifoil R2/2 holey carbon copper grids (Quantifoil Micro Tools, Germany). The grids were blotted for 4 s, blot force −2, at 100% humidity, 10 °C and plunge frozen in liquid ethane using an FEI Vitrobot Mark IV. Micrographs were recorded on FEI Titan Krios microscopes operating at 300 kV at a nominal magnification of 130,000 for U-BME and 160,000 for U-SHA with calibrated pixel sizes of 1.07 and 0.86 Å, respectively. The microscopes were equipped with Gatan K2 cameras. Movies were recorded using the EPU control software and videos were collected for 6 s with a total dose of 41.2 e$^-$/Å$^2$ and a calibrated dose of 1.03 e$^-$/Å$^2$ per fraction in a total of 40 fractions at defocus values between −0.5 and −3 µm.

**Image processing.** Sets of 900 and 1890 micrographs were collected for U-BME and U-SHA, respectively. Image processing was carried out using the program Relion-3.1-beta[24]. Image drift correction was performed using Relion's own implementation and whole micrograph CTF estimation was performed using the program CTFFIND4.1.10 in the Relion workflow[43]. The initial datasets contained 193,289 particles and 276,994 particles for U-BME and U-SHA, respectively. Reference-free 2D classification resulted in two datasets with 175,895 and 187,461 particles, respectively. 3D classification resulted in one main class. 3D refinement applying tetrahedral symmetry (Supplementary Fig. 6) was performed based on a 60 Å low pass-filtered map of dodecameric urease generated from the previously published 3 Å resolution native crystal structure (PDB id: 1E9Z)[23]. Per-particle CTF refinement and Bayesian polishing were performed iteratively and the best 3D classes were selected for further processing (Supplementary Fig. 2). The program Resolve of the Phenix package (version 1.18.1-3865) was used for map density modification[25]. The program UCSF Chimera was used for visualization of cryo-EM maps and fitting of atomic models[44]. Figures were generated with UCSF Chimera. The program Coot was used for rebuilding, evaluating and comparing models and maps and Phenix was used for model real space refinement including Ramachandran restraints[27,45].

**Data deposition.** The cryo-EM maps were deposited in the Electron Microscopy Data Bank with accession codes EMD-4629 and EMD-11233 for the cryo-EM maps of U-BME and the U-SHA, respectively. Atomic model coordinates of the refined complexes were deposited in the Protein Data Bank with accession numbers 6QSU and 6ZJA for U-BME and U-SHA, respectively.

**Urease inhibition assay.** A phenol red spectrophotometric assay (absorption at 562 nm) measuring pH increase was performed with purified *H. pylori* urease in vitro as previously reported[29]. Different concentrations of *H. pylori* urease from 4.8 ng/mL to 10 µg/mL were prepared in assay buffer (2 mM potassium phosphate pH 6.8 and 0.1 mM EDTA). Fifty microliters of each of the urease concentrations were added to wells in a 96-well plate. Fifty microliters of phenol red solution containing 35 µg/mL phenol red, 3 mM sodium phosphate pH 6.8 and 6 mM urea were added to the wells immediately after. The plate was incubated for 1 h at 30 °C in an Elx808 plate reader (BioTek Instruments), recording the absorption at 562 nm every 20 s. Optimal urease concentration was determined to be 800 ng/mL, where the absorption curve was linear and the increase in optical density reached the maximum velocity after 40 min. For the assay, 20x stock solutions of dimethyl sulfoxide (DMSO)-dissolved inhibitors were individually added to the 800 ng/mL urease sample in assay buffer to obtain final inhibitor concentrations ranging from 400 nM to 200 mM for BME and from 2 nM to 1 mM for SHA, and mixed well. DMSO alone was used as the control and assay buffer without protein as the blank. Fifty microliters of each of the inhibitor concentrations mixed with protein were applied to a 96-well plate in triplicates. Urea hydrolysis was initiated by adding 50 µL of phenol red solution containing 6 mM urea to each well, causing the absorbance at 562 nm to increase due to the increase of the pH during the reaction caused by ammonia production. The plates were incubated as described above. The absorption values at the time were the increase of absorption at 562 nm reached its maximum velocity were plotted against inhibitor concentrations and IC$_{50}$ values were calculated using Gen5 software (BioTek Instruments).

**Reporting summary**. Further information on research design is available in the Nature Research Reporting Summary linked to this article.

## Data availability

Data supporting the findings of this manuscript are available from the corresponding authors upon reasonable request. A reporting summary for this article is available as a Supplementary Information file. The Cryo-EM density maps along with half maps and corresponding masks have been deposited in the Electron Microscopy Data Bank (https://www.ebi.ac.uk/pdbe/emdb) under accession codes: EMD-4629 (U-BME complex) and EMD-11233 (U-SHA complex). The atomic coordinates have been deposited to the RCSB Protein Data Bank (https://www.rcsb.org/) under codes 6QSU (U-BME complex) and 6ZJA (U-SHA complex). The source data underlying Fig. 3 are provided as an excel file. Source data are provided with this paper.

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

## Acknowledgements

This paper is dedicated to the memory of Deryck J. Mills, who was a colleague, collaborator, friend, and mentor with enormous enthusiasm, generosity to teach and love of science, shaping the careers of scientists all over the world. We thank Michael Hall and Linda Sandblad for help with cryo-EM data collection, Tom Terwilliger for help with Resolve density modification, and Janet Vonck for suggestions regarding data processing and reviewing the manuscript. We thank Dr Eric Samuels for initial contributions to the urease purification protocol and activity assay. The data for U-SHA was collected at the Umeå Core Facility for Electron Microscopy, a node of the Cryo-EM Swedish National Facility, funded by the Knut and Alice Wallenberg, Family Erling Persson and Kempe Foundations, SciLifeLab, Stockholm University and Umeå University. The authors acknowledge Research Council of Norway grant number 275207 for funding towards running costs and positions. E.S.C. is also supported by an H2020 MSCA Individual Fellowship 795980, M.S.G. by an H2020 MSCA Scientia II fellowship, and H.L. by NCMM core funding (Research Council of Norway grant number 187615 and South-Eastern Norway Regional Health Authority). The research leading to these results has received funding from the European Union's Horizon 2020 research and innovation programme under the Marie Skłodowska-Curie grant agreements No. 795980 and No. 801133.

## Author contributions

E.S.C. prepared the grids, collected the cryo-EM data, performed the data processing and the 3D reconstruction, built the molecular model, and contributed to interpretation. X.C. cloned and purified the protein, and performed initial inhibition assays. M.S.G. contributed to the inhibition assays. D.J.M. contributed to the grid preparation, and the data collection. H.L. contributed to data collection, processing, model building, and interpretation. All authors contributed to the study design and to writing the manuscript.

## Competing interests

The authors declare no competing interests.
