## [Peer Review File · Nature Communications]

REVIEWER COMMENTS

Reviewer #1 (Remarks to the Author):

The paper by Hartmut Luecke and co-workers describes the obtainment of a high resolution (2.01 Å) structure of urease from *Helicobacter pylori* using cryo-EM. The main conclusions of the work are that cryo-EM can be used to obtain structural data for ureases from bacterial human pathogens, valuable for the development of inhibitors with high specificity and affinity. Moreover, the paper describes a comparison of two cryo-EM structures of *H. pylori* urease complexed with a novel inhibitor (2- $\{[1-(3,5\text{-dimethylphenyl})\text{-}1\text{H-imidazol-}2\text{-yl]sulfanyl}\}$ -N-hydroxyacetamide) or with beta-mercaptoethanol, a molecule known to be a urease inhibitors since many years. The paper should be published with minor revisions as I indicate below, to improve the manuscript impact:

In general, the paper does not follow a clear outline and that should be corrected. Topics are discussed without following a clear line of thought. Please revise this aspect.

Line 32-33

While it is true that in *H. pylori* urease is produced in the bacterium cytoplasm (where else should it be produced?) it should be specified that it is the extra-cytoplasmic form that is critical for the establishment of a near neutral pH around the bacterial cells.

Line 75-77

It should be specified that the first enzyme to be crystallized was urease from *Canavalia ensiformis* (jack bean), while the first urease crystal structure belonged to the bacterium *Klebsiella aerogenes*.

Line 79-80

It has been widely established that the ligand bridging the two Ni(II) ions in the active site of urease is a hydroxide anion. Thus, this is not "thought". Please add the following references that support this information:

Benini, S.; Cianci, M.; Mazzei, L.; Ciurli, S. *J Biol Inorg Chem* 2014, 19 (8), 1243–1261.

Benini, S.; Rypniewski, W. R.; Wilson, K. S.; Miletti, S.; Ciurli, S.; Mangani, S. *Structure* 1999, 7 (2), 205–216.

Line 81-86

The review by Kafarsky should be complemented by the following reviews:

Mazzei, L.; Musiani, F.; Ciurli, S. *J Biol Inorg Chem* 2020, 25 (6), 829–845.

Mazzei, L.; Musiani, F.; Ciurli, S. Zamble, D., Rowinska-Zyrek, M., Kozłowski, H., Eds.; *Royal Society of Chemistry: Cambridge*, 2017; pp 60–97.

Maroney, M. J.; Ciurli, S. *Chem Rev* 2014, 114 (8), 4206–4228.

Benini, S.; Musiani, F.; Ciurli, S. *Encyclopedia of Metalloproteins* 2013, 2287–2292.

Zambelli, B.; Musiani, F.; Benini, S.; Ciurli, S. *Acc Chem Res* 2011, 44 (7), 520–530.

Lines 102-104

In the main text the authors say: "We report cryo-EM maps of *H. pylori* urease at 2.45 Å and 2.01 Å resolution, the highest resolution to date for *H. pylori* urease, of sufficient detail to aid in drug development (Fig. 1 and Supplementary Fig. 1 and 2)". However, Fig.1 and 1SI show the Cryo-EM density map at 2.01 Å resolution of dodecameric 1.1 MDa *Helicobacter pylori* urease with bound SHA, while Fig. 2SI describes the data collection and processing procedure. No pictures of the cryo-EM maps of *H. pylori* urease at 2.45 Å resolution are shown. The authors should provide them.

Line 111

Supplementary Fig.1 shows only U-SHA. Either change the sentence or the figure.

Lines 130-131

Here the authors talk about bacterial ureases. In fact, plant ureases are dimers of trimers.

Lines 144-149

Here the authors include a part describing kinetic characterization of the inhibition in a mainly structural paragraph. This should be corrected.

Lines 158-159

Papers more recent than Pearson (2000) should be cited since they give a more complete interpretation of flap properties. The authors should include especially the following citations: Benini, S.; Rypniewski, W. R.; Wilson, K. S.; Miletti, S.; Ciurli, S.; Mangani, S. *Structure* 1999, 7 (2), 205–216.

Musiani, F.; Arnofi, E.; Casadio, R.; Ciurli, S. *J Biol Inorg Chem* 2001, 6 (3), 300–314.

Mazzei, L.; Cianci, M.; Benini, S.; Ciurli, S. *Chemistry* 2019, 25 (52), 12145–12158.

Lines 160-162

The sentence "Interestingly, the flap region shows the highest RMSD backbone variations of both UreA and UreB subunits (above 1.5 Å), whereas the active site residues nearby exhibit very low RMSD between U-BME and U-SHA (Fig. 4B)." hints to a surprise by the authors, while it has been already established in a large number of papers (essentially in all papers reporting urease structures) that the urease active site is rigid, while the flap covering the active site is mobile. See comment above.

Line 163-164

In SPU, actually two molecules of BME were found in the active site cavity, one directly binding the Ni ions and the other one covalently bound to the Cys322 residue (SPU numbering). The authors should specify this point and indicate whether they see a similar situation in their structure.

Line 192

The authors should use the term "apo" only to indicate, if necessary, the urease enzyme devoid of the essential Ni(II) ions, and indicate as "native" the form of the enzyme devoid of bound inhibitors.

Lines 219-220

The authors should indicate that hydroxamic acids cannot be used for medical purposes.

Lines 232-239

The following citations should be added:

Benini, S.; Rypniewski, W. R.; Wilson, K. S.; Miletti, S.; Ciurli, S.; Mangani, S. *Structure* 1999, 7 (2), 205–216.

Musiani, F.; Arnofi, E.; Casadio, R.; Ciurli, S. *J Biol Inorg Chem* 2001, 6 (3), 300–314.

Mazzei, L.; Cianci, M.; Benini, S.; Ciurli, S. *Chemistry* 2019, 25 (52), 12145–12158.

Reviewer #2 (Remarks to the Author):

This is a good paper about determining the structure of an enzyme with a drug bound that has high clinical relevance. The structure is important and should be published at once after the problems are fixed. Fortunately, none of the problems are fundamental to the paper or its findings. They are really just a case of overenthusiastic use of the data processing programs without a deep understanding of how this might cause artefacts. They can easily be remedied with a bit of reprocessing of the data with more caution.

1. The authors have performed 'density modification' of the cryoEM map which is dangerous at best and makes it impossible to determine if they have over-interpreted their data. The FSCs in Sfig 2 show a sharp drop which indicates a severe processing defect of some sort. There is also aliasing in Sfig 2a

and the density differences in Sfig 1 are not believable without seeing the mask and the surrounding density. To sort this out:

- a. The authors should calculate the FSC's between unfiltered half maps, and with high resolution phase randomization (both are standard outputs of Relion which they likely already have).
- b. They should calculate the FSC between the model and the non-density modified map that is the output of Relion and report this in the supplement as well.
- c. They should plot the particle orientation distribution and evaluate it for coverage of Fourier space.
- d. They should check their CTF fits for aliasing artefacts and increase the sampling/padding if required.

2. The claims about structure based drug design need to be removed since they have not actually done this in this study. What they have done is solved the structure of an enzyme with a drug bound that was found in a high throughput screen. Many other cryoEM studies have shown drugs bound to complexes and there are no new methods in this paper. This does not take away from the quality or importance of the work, the authors should just tone down the hyperbole, in particular:

- a. L220-221 phrase about "that opens the door...EM" is unsupported and should be deleted.
- b. Delete the paragraph from L240-248, just waffle that says nothing interesting.

3. For resolution reports in the text, use 2 significant digits

4. Ramachandran plots: On line 119 the authors seem to claim that their models being free of Ramachandran outliers means their maps are better. This is not correct- they have used Ramachandran restraints in Phenix which forces the model to have only allowed angles regardless if the data supports it. The models should be refined without Ramachandran restraints and all outlying rotamers carefully compared and deposited (the ones on line 124) in the pdb if they wish to make this comparison. Otherwise just delete it completely and be sure to specify in the methods the type of restraints that were imposed during model refinement.

Other minor edits

Fig 2 the data points are hard to see and interpret

L33 2.01 -> 2.0

L35 2.45 -> 2.5

L47 Current eradication treatment, called ... entails ingesting a ...

L48 ... antibiotics clarithromycin and metronidazole. Resistance ...

L 102-111 2 sig. figs on all the resolutions please.

Paragraph on L138-141 should be moved to the discussion

Paragraph on L212-218 is redundant with intro - delete

Reviewer #1:

The paper by Hartmut Luecke and co-workers describes the obtainment of a high resolution (2.01 Å) structure of urease from *Helicobacter pylori* using cryo-EM. The main conclusions of the work are that cryo-EM can be used to obtain structural data for ureases from bacterial human pathogens, valuable for the development of inhibitors with high specificity and affinity. Moreover, the paper describes a comparison of two cryo-EM structures of *H. pylori* urease complexed with a novel inhibitor (2-{{1-(3,5-dimethylphenyl)-1H-imidazol-2-yl}sulfanyl}-N-hydroxyacetamide) or with beta-mercaptoethanol, a molecule known to be a urease inhibitors since many years. The paper should be published with minor revisions as I indicate below, to improve the manuscript impact:

In general, the paper does not follow a clear outline and that should be corrected. Topics are discussed without following a clear line of thought. Please revise this aspect.

Line 32-33

While it is true that in *H. pylori* urease is produced in the bacterium cytoplasm (where else should it be produced?) it should be specified that it is the extra-cytoplasmic form that is critical for the establishment of a near neutral pH around the bacterial cells.

We thank the reviewer for the comment, however the mention of “urease in the cytoplasm” in the abstract is not meant to state where urease is produced (indeed by ribosomes in the cytoplasm), but where the bulk of the acid-sensitive urease is located and where substrate (urea) arrives through the plasma membrane urea channel, UreI. We are aware of an unresolved controversy about the role of extracellular urease, but this is not the focus of this paper.

Line 75-77

It should be specified that the first enzyme to be crystallized was urease from *Canavalia ensiformis* (jack bean), while the first urease crystal structure belonged to the bacterium *Klebsiella aerogenes*.

We have now clarified this in the main text.

Line 79-80

It has been widely established that the ligand bridging the two Ni(II) ions in the active site of urease is a hydroxide anion. Thus, this is not “thought”. Please add the following references that support this information:

Benini, S.; Cianci, M.; Mazzei, L.; Ciurli, S. *J Biol Inorg Chem* 2014, 19 (8), 1243–1261.

Benini, S.; Rypniewski, W. R.; Wilson, K. S.; Miletti, S.; Ciurli, S.; Mangani, S. *Structure* 1999, 7 (2), 205–216.

We have altered “is thought” to “has been reported” and added the citation where the hydroxide anion mechanism is described: *Benini, S.; Rypniewski, W. R.; Wilson, K. S.; Miletti, S.; Ciurli, S.; Mangani, S. Structure* 1999, 7 (2), 205–216. “A new proposal for urease mechanism based on the crystal structures of the native and inhibited enzyme from *Bacillus pasteurii*: why urea hydrolysis costs two nickels”.

Line 81-86

The review by Kafarsky should be complemented by the following reviews:

Mazzei, L.; Musiani, F.; Ciurli, S. *J Biol Inorg Chem* 2020, 25 (6), 829–845.

Mazzei, L.; Musiani, F.; Ciurli, S. Zamble, D., Rowinska-Zyrek, M., Kozłowski, H., Eds.; Royal Society of Chemistry: Cambridge, 2017; pp 60–97.

Maroney, M. J.; Ciurli, S. *Chem Rev* 2014, 114 (8), 4206–4228.

Benini, S.; Musiani, F.; Ciurli, S. *Encyclopedia of Metalloproteins* 2013, 2287–2292.

Zambelli, B.; Musiani, F.; Benini, S.; Ciurli, S. *Acc Chem Res* 2011, 44 (7), 520–530.

We are sorry for the oversight, we have added the 2020 review by Ciurli et al. This review points also to some of the other, less recent reviews mentioned above. *L. Mazzei, F. Musiani and S. Ciurli "The*

structure-based reaction mechanism of urease, a nickel dependent enzyme: tale of a long debate" J. Biol. Inorg. Chem. (2020) 25:829-845.

Lines 102-104

In the main text the authors say: “We report cryo-EM maps of *H. pylori* urease at 2.45 Å and 2.01 Å resolution, the highest resolution to date for *H. pylori* urease, of sufficient detail to aid in drug development (Fig. 1 and Supplementary Fig. 1 and 2)”. However, Fig.1 and 1SI show the Cryo-EM density map at 2.01 Å resolution of dodecameric 1.1 MDa *Helicobacter pylori* urease with bound SHA, while Fig. 2SI describes the data collection and processing procedure. No pictures of the cryo-EM maps of *H. pylori* urease at 2.45 Å resolution are shown. The authors should provide them.

We have added cryo-EM density for U-BME to the Supplemental Information (Fig. 1).

Line 111

Supplementary Fig.1 shows only U-SHA. Either change the sentence or the figure.

We have changed the figure and the sentence.

Lines 130-131

Here the authors talk about bacterial ureases. In fact, plant ureases are dimers of trimers.

We have added “dimers of trimers” to the sentence.

Lines 144-149

Here the authors include a part describing kinetic characterization of the inhibition in a mainly structural paragraph. This should be corrected.

Our intent in measuring and providing IC50 values for the two inhibitors that we also analyze structurally is to highlight the IC50 difference of nearly 3 orders of magnitude. We could have chosen to provide this information after the presentation of structural data, however, we feel it is preferable for the reader to know this while reading the structural section of our results.

Lines 158-159

Papers more recent than Pearson (2000) should be cited since they give a more complete interpretation of flap properties. The authors should include especially the following citations: Benini, S.; Rypniewski, W. R.; Wilson, K. S.; Miletti, S.; Ciurli, S.; Mangani, S. *Structure* 1999, 7 (2), 205–216.

Musiani, F.; Arnofi, E.; Casadio, R.; Ciurli, S. *J Biol Inorg Chem* 2001, 6 (3), 300–314.

Mazzei, L.; Cianci, M.; Benini, S.; Ciurli, S. *Chemistry* 2019, 25 (52), 12145–12158.

We thank the reviewer for pointing out the more recent paper, we have included the 2019 reference by Mazzei *et al.* “*The Impact of pH on Catalytically Critical Protein Conformational Changes: The Case of the Urease, a Nickel Enzyme*”. The other two references are from the same corresponding author and are cited in the 2019 reference.

Lines 160-162

The sentence “Interestingly, the flap region shows the highest RMSD backbone variations of both UreA and UreB subunits (above 1.5 Å), whereas the active site residues nearby exhibit very low RMSD between U-BME and U-SHA (Fig. 4B).” hints to a surprise by the authors, while it has been already established in a large number of papers (essentially in all papers reporting urease structures) that the urease active site is rigid, while the flap covering the active site is mobile. See comment above.

We have removed the word “interestingly”.

Line 163-164

In SPU, actually two molecules of BME were found in the active site cavity, one directly binding the Ni ions and the other one covalently bound to the Cys322 residue (SPU numbering). The authors should specify this point and indicate whether they see a similar situation in their structure.

We do not observe density for a second BME near Cys321 (*H. pylori* numbering) at the tip of the flap in U-BME. We have added a sentence clarifying this point.

Line 192

The authors should use the term “apo” only to indicate, if necessary, the urease enzyme devoid of the essential Ni(II) ions, and indicate as “native” the form of the enzyme devoid of bound inhibitors.

This is an excellent point and we have changed all references to “apo” (which would imply the absence of cofactors such as nickel) to “native”.

Lines 219-220

The authors should indicate that hydroxamic acids cannot be used for medical purposes.

While there are known issues with hydroxamic acids as therapeutics (*ie Graham F. Smith, in Progress in Medicinal Chemistry, 2011*), at least one hydroxamic acid has been approved by the U.S. FDA for human use: <https://en.wikipedia.org/wiki/Vorinostat>

Lines 232-239

The following citations should be added:

Benini, S.; Rypniewski, W. R.; Wilson, K. S.; Miletti, S.; Ciurli, S.; Mangani, S. Structure 1999, 7 (2), 205–216.

Musiani, F.; Arnofi, E.; Casadio, R.; Ciurli, S. J Biol Inorg Chem 2001, 6 (3), 300–314.

Mazzei, L.; Cianci, M.; Benini, S.; Ciurli, S. Chemistry 2019, 25 (52), 12145–12158.

This discussion is specifically about substrate access to the active site in the spherical dodecameric arrangement of the *H. pylori* enzyme. We have made this more clear by adding the term dodecamer early in the first sentence.

Reviewer #2:

1. The authors have performed 'density modification' of the cryoEM map which is dangerous at best and makes it impossible to determine if they have over-interpreted their data. The FSCs in Sfig 2 show a sharp drop which indicates a severe processing defect of some sort. There is also aliasing in Sfig 2a and the density differences in Sfig 1 are not believable without seeing the mask and the surrounding density. To sort this out:

a. The authors should calculate the FSC's between unfiltered half maps, and with high resolution phase randomization (both are standard outputs of Relion which they likely already have).

We thank the reviewer for the suggestions and comments. The recently published procedure of EM map density modification (Terwilliger *et al.*, 2020) was not essential for our analysis as we obtained only modest improvements of nominal resolution and map quality. We have indeed analyzed the map-to-map FSCs between unfiltered half maps (generated by Relion with high resolution phase randomization before density modification) and this results in resolution estimates of 2.09 Å for U-SHA and of 2.55 Å for U-BME, respectively, as stated in the manuscript. The corresponding FCSs do not show a sharp drop which means that masking effects are not present. We have now added these FSC plots to Supplementary Fig. 2. It is only after the density modification (as described in the methods paper by Terwilliger *et al.*) that we get modest resolution estimation improvements of 0.08 Å and a 0.1 Å for U-SHA and U-BME, respectively. The sharp drop is only observed in the density modified FSC curves and corresponds to a resolution of twice the pixel size, which naturally is beyond the resolution limit we are considering.

b. They should calculate the FSC between the model and the non-density modified map that is the output of Relion and report this in the supplement as well.

We had determined the model-to-map FSC between the Relion maps that agree well with the FSC calculated from the half maps. We have added the Relion map-to-model FSCs to Supplementary Fig. 2.

c. They should plot the particle orientation distribution and evaluate it for coverage of Fourier space.

This plot has been added as Supplementary Fig. 3, however we would like to note that we do have tetrahedral symmetry (7), which significantly reduces unique orientational space.

d. They should check their CTF fits for aliasing artefacts and increase the sampling/padding if required.

The reviewer is correct in pointing out the CTF aliasing issue in the micrograph originally pictured in Supplementary Fig. 2. We had picked this micrograph of the lower-resolution U-BME data set because it showed the particles nicely due to a high defocus value. In general, only high-defocus micrographs which contribute little to the density of U-BME, displayed aliasing. We have re-run CTF estimation with CTFFind 4.1.10 using 2048-pixel box size for the amplitude spectrum instead of the default of 512 pixels, removing the aliasing as shown in updated Supplementary Fig 2. Incidentally, the larger box size does not change the parameters determined by CTFFind 4.1.10 by more than 0.004%. Furthermore, we do not observe any CTF aliasing for the U-SHA high-resolution data set which was recorded with a smaller pixel size. Importantly, for both data sets, as described in the methods section, we used two rounds of the subsequent per-particle CTF refinement option of Relion-3 during the later stages of refinement. We have clarified this in the methods section and we have added a panel to Sup. Fig. 2 showing the gradient of per-particle defocus values within one representative micrograph. Here is the citation and an excerpt from the paper describing per-particle CTF refinement:

Zivanov J, Nakane T, Forsberg BO, Kimanius D, Hagen WJ, Lindahl E, Scheres SH. New tools for automated high-resolution cryo-EM structure determination in RELION-3. *Elife*. 2018 Nov 9;7:e42166. doi: 10.7554/eLife.42166. PMID: 30412051; PMCID: PMC6250425.

CTF refinement

In RELION, per-micrograph CTF parameters are determined through wrappers to CTFFIND (Rohou and Grigorieff, 2015) or Gctf (Zhang, 2016). These programs fit CTFs to the Thon rings visible in the power spectra of (patches of) micrographs. In RELION-3, we have implemented a program to refine the CTF parameters, that is to re-estimate defocus and astigmatism, using a 3D reference structure. This allows CTF estimation to exploit both the phases and the amplitudes of the experimental images, instead of having to rely exclusively on their power spectra. This approach thus produces significantly more reliable estimates, due to higher signal-to-noise ratios and also because it does not require separation of the Thon rings from the background intensity of the power spectrum. This is illustrated in Figure 1. The increased stability of the CTF fitting can be leveraged to estimate independent defoci for individual particles. Similar functionality also exists in Frealign (Grigorieff, 2007) and cisTEM (Grant et al., 2018).

2. The claims about structure based drug design need to be removed since they have not actually done this in this study. What they have done is solved the structure of an enzyme with a drug bound that was found in a high throughput screen. Many other cryoEM studies have shown drugs bound to complexes and there are no new methods in this paper. This does not take away from the quality or importance of

the work, the authors should just tone down the hyperbole, in particular:

a. L220-221 phrase about "that opens the door...EM" is unsupported and should be deleted.

We did not intend to state that we have carried out structure based drug design but that our work paves a way for doing so in the future. We have corrected the sentence so that our intention becomes clearer.

b. Delete the paragraph from L240-248, just waffle that says nothing interesting.

In this paragraph, which is at the very end of the article, our aim is to provide an outlook of the challenges and developments of cryo-EM. We would prefer to keep this outlook and would like the editor to make the call if it needs to be deleted.

3. For resolution reports in the text, use 2 significant digits

This has been corrected.

4. Ramachandran plots: On line 119 the authors seem to claim that their models being free of Ramachandran outliers means their maps are better. This is not correct- they have used Ramachandran restraints in Phenix which forces the model to have only allowed angles regardless if the data supports it. The models should be refined without Ramachandran restraints and all outlying rotamers carefully compared and deposited (the ones on line 124) in the pdb if they wish to make this comparison. Otherwise just delete it completely and be sure to specify in the methods the type of restraints that were imposed during model refinement.

Ramachandran restraints will not necessarily remove all Ramachandran outliers from a model as can be seen in our re-refinement of the two 3.0 Å *H. pylori* urease crystal structures published in 2001 using the same Ramachandran restraints as we used for the refinement of our models with our cryo EM maps. For example, the 2001 apo crystal structure shows 7.1% Ramachandran outliers, after re-refinement with Ramachandran restraints, that percentage is reduced by less than 50% to 3.6%. We are stating now that we used Phenix Ramachandran restraints for refinements using both cryo EM as well as electron densities.

Other minor edits

Fig 2 the data points are hard to see and interpret

We suspect the reviewer is referring to Fig. 3 of the manuscript (IC50 plots). We have replotted the data for more clarity.

L33 2.01 -> 2.0

Corrected.

L35 2.45 -> 2.5

Corrected.

L47 Current eradication treatment, called ... entails ingesting a ...

Corrected.

L48 ... antibiotics clarithromycin and metronidazole. Resistance ...

Corrected.

L 102-111 2 sig. figs on all the resolutions please.

Corrected.

Paragraph on L138-141 should be moved to the discussion

While we agree that this section is more of a discussion topic, we feel it would be out of context and too detailed for our short general discussion. We are happy to have the editor guide us on this issue.

Paragraph on L212-218 is redundant with intro – delete

This is the start of our discussion and we feel that reminding the reader of the big picture is important. We would like to keep it and would like the editor to make the call if it should or not be deleted.

Reviewer's Comments:

Reviewer #1 (Remarks to the Author)

The authors have satisfied essentially all my comments and suggestions. I find the paper now suitable for publication, and compliments for the nice work!

Reviewer #2 (Remarks to the Author)

2nd review of NCOMMS-20-35875A

The current manuscript is much improved over the previous version. The revision is applauded and the paper is going to be well placed in the field. There are a few remaining problems to address. To remedy these with a minimum of effort, I suggest the authors do the following:

1. Delete Supplementary figure 1

This comparison is uninformative at best. Anyone can sharpen a map to make it look like there are holes in aromatics. Just remove it. Everyone will be able to look at the final deposited maps to judge the quality of the reconstructions.

2. Supplementary figure 2, which is crucial to evaluating the structure, is now much more informative, but still has errors.

Panels d) and e) show the FSC plots of interest now, but still do not include the high resolution noise substituted / phase randomised FSC plots. I suggest the authors reread S. Chen et al. / Ultramicroscopy 135 (2013) p24–35 and then do the plot again with the data that is already output from Relion. It should include FSCs of 1. unmasked half map, 2. phase randomized, masked half-maps, 3. map vs model. The density modified FSCs should be kept in separate panels as they have done.

3. The resolution values reported in the paper and deposited in the PDB / EMDB should be the ones from Sfig 2 d) & e), namely 2.6 and 2.1 Å respectively.

The unfiltered, unmodified half maps and the mask should be deposited in the EMDB, along with the final sharpened, masked map used for modelling, and the density modified maps. *All* of these should be deposited in the EMDB entry.

Reviewer #1 (Remarks to the Author)

The authors have satisfied essentially all my comments and suggestions. I find the paper now suitable for publication, and compliments for the nice work!

We thank reviewer #1 for the comments and for helping to improve our manuscript.

Reviewer #2 (Remarks to the Author)

2nd review of NCOMMS-20-35875A

The current manuscript is much improved over the previous version. The revision is applauded and the paper is going to be well placed in the field. There are a few remaining problems to address. To remedy these with a minimum of effort, I suggest the authors do the following:

We thank reviewer #2 for the comments and their input on improving our manuscript.

1. Delete Supplementary figure 1

This comparison is uninformative at best. Anyone can sharpen a map to make it look like there are holes in aromatics. Just remove it. Everyone will be able to look at the final deposited maps to judge the quality of the reconstructions.

We originally showed a comparison of U-SHA density only before and after density modification as it does show modest differences in the map quality and in this revised version we have added the U-BME density as requested by reviewer #1. While we agree that the differences in density for the U-BME are not visible in the selected stretch of residues, there are visible differences for the U-SHA density. We also believe that not everyone who will read the manuscript will be structural biologists and would download the maps to judge quality of the reconstructions. We therefore believe that figure 1 should be kept, however we are happy to have the editor make this decision.

2. Supplementary figure 2, which is crucial to evaluating the structure, is now much more informative, but still has errors.

Panels d) and e) show the FSC plots of interest now, but still do not include the high resolution noise substituted / phase randomised FSC plots. I suggest the authors reread S. Chen et al. / Ultramicroscopy 135 (2013) p24–35 and then do the plot again with the data that is already output from Relion. It should include FSCs of 1. unmasked half map, 2. phase randomized, masked half-maps, 3. map vs model. The density modified FSCs should be kept in separate panels as they have done.

The requested curve has been added.

3. The resolution values reported in the paper and deposited in the PDB / EMDB should be the ones from Sfig 2 d) & e), namely 2.6 and 2.1 Å respectively.

We discussed our density modification procedure extensively with the first author of the published method (Terwilliger T. *et al*, 2020 *Nature Methods* volume 17, pp 923–927) and we have used the density modified map for model refinement and model building, therefore we think it represents best our procedure to report the resolution after the density modification rather than before. In any case, the nominal resolution improvement due to density modification is marginal (0.09 Å) and not essential in any way to support our findings.

The unfiltered, unmodified half maps and the mask should be deposited in the EMDB, along with the final sharpened, masked map used for modelling, and the density modified maps. *All* of these should be deposited in the EMDB entry.

We have deposited the density-modified maps in the current entry and we have deposited now the remaining maps and masks.